# Galactose in the Post-Weaning Diet Programs Improved Circulating Adiponectin Concentrations and Skeletal Muscle Insulin Signaling

**DOI:** 10.3390/ijms231810207

**Published:** 2022-09-06

**Authors:** Peixin Sun, Lianne M. S. Bouwman, Jo-lene de Deugd, Inge van der Stelt, Annemarie Oosting, Jaap Keijer, Evert M. van Schothorst

**Affiliations:** 1Human and Animal Physiology, Wageningen University and Research, 6708 WD Wageningen, The Netherlands; 2Danone Nutricia Research, 3584 CT Utrecht, The Netherlands

**Keywords:** nutritional programming, galactose, insulin resistance, adipose tissue, adiponectin, inflammation

## Abstract

Short-term post-weaning nutrition can result in long-lasting effects in later life. Partial replacement of glucose by galactose in the post-weaning diet showed direct effects on liver inflammation. Here, we examined this program on body weight, body composition, and insulin sensitivity at the adult age. Three-week-old female C57BL/6JRccHsd mice were fed a diet with glucose plus galactose (GAL; 16 energy% (en%) each) or a control diet with glucose (GLU; 32 en%) for three weeks, and afterward, both groups were given the same high-fat diet (HFD). After five weeks on a HFD, an oral glucose tolerance test was performed. After nine weeks on a HFD, energy metabolism was assessed by indirect calorimetry, and fasted mice were sacrificed fifteen minutes after a glucose bolus, followed by serum and tissue analyses. Body weight and body composition were not different between the post-weaning dietary groups, during the post-weaning period, or the HFD period. Glucose tolerance and energy metabolism in adulthood were not affected by the post-weaning diet. Serum adiponectin concentrations were significantly higher (*p* = 0.02) in GAL mice while insulin, leptin, and insulin-like growth factor 1 concentrations were not affected. Expression of *Adipoq* mRNA was significantly higher in gonadal white adipose tissue (gWAT; *p* = 0.03), while its receptors in the liver and skeletal muscles remained unaffected. *Irs2* expression was significantly lower in skeletal muscles (*p* = 0.01), but not in gWAT or *Irs1* expression (in both tissues). Gene expressions of inflammatory markers in gWAT and the liver were also not affected. Conclusively, galactose in the post-weaning diet significantly improved circulating adiponectin concentrations and reduced skeletal muscle *Irs2* expression in adulthood without alterations in fat mass, glucose tolerance, and inflammation.

## 1. Introduction

Obesity is a disease characterized by increased white adipose tissue (WAT) mass and subsequent WAT dysfunction [1], which (in a large majority of cases) causes adverse metabolic consequences, such as insulin resistance [2].

Early life, ranging from conception, pregnancy, and lactation into the post-weaning period is critical for development. Nutrition in early life can result in long-lasting metabolic health effects in adulthood, a process referred to as nutritional programming [3], which has been shown to affect obesity and associated metabolic diseases [4,5]. With the increased prevalence of obesity, interest in the development of preventive strategies has also increased. Thus, full understanding of the early life conditions that influence obesity and associated metabolic disease is essential.

Nutritional programming during pregnancy and lactation is well established in animal models. Both beneficial and adverse effects on later life metabolic health can occur, depending on the timing of the exposure and the type and amount of nutrients, often with sex-specific effects. For example, mild calorie restriction of rat dams during pregnancy increased the risk for the development of obesity in their offspring, while mild calorie restriction during lactation gave a protective effect [6,7]. A limited number of studies focused on nutritional interventions during the post-weaning period. Nevertheless, nutritional programming can also be achieved in the immediate post-weaning period, as it has been demonstrated especially for exposure to dietary lipids. Multiple rodent studies have shown that the dietary lipid structure in the post-weaning period given for a brief period can beneficially program metabolic health in later life [8,9,10,11,12]. Likewise, medium-chain fatty acids in the early post-weaning diet program resulted in lower adiposity with smaller adipocytes in later life in male mice, with a tendency for improved glucose tolerance [13]. Continuous feeding of both pre- and probiotics during lactation and early post-weaning program resulted in lower fat mass accumulation in adulthood, with improved insulin sensitivity [14]. Although metabolic programming by dietary lipids given within the post-weaning period is shown convincingly, programming by post-weaning carbohydrates is much less established.

Previously, we investigated to what extent different carbohydrate fractions of the post-weaning diet can program metabolic health in an adult obesogenic environment. A lowly- versus highly-digestible starch diet fed in the post-weaning period affected metabolic flexibility beneficially in high-fat diet (HFD)-challenged female mice [15]. A post-weaning isocaloric diet with 32 energy% (en%) fructose instead of glucose reduced serum insulin concentrations and homeostatic model assessment for the insulin resistance (HOMA-IR) index in adult HFD-fed female mice, without significant changes in body weight or fat mass [16]. In addition, we showed that replacing part of glucose with galactose in the post-weaning diet, to mimic an extended exposure duration to lactose present in milk, gave the most striking effects: it reduced body weight, fat mass, and circulating insulin levels in adult HFD-fed female mice [17]. Of note, the monosaccharide galactose content represented only a small fraction of the total carbohydrates in the diet. After just three weeks, post-weaning dietary intervention direct effects were also observed in young female mice, including reduced hepatic triglycerides levels, reduced hepatic expression of inflammatory markers, and reduced systemic markers of inflammation in the liver and circulation [18]. Therefore, galactose seems to have beneficial effects both directly as well as via nutritional programming.

As the studies focusing on galactose-induced metabolic programming effects are still very scarce, we questioned whether the previously observed effects [17] are robust, e.g., independent of dietary background during the prenatal phase or the dietary composition of the obesogenic HFD during adulthood. Therefore, a metabolic programming study was performed by feeding dams normal chow during pregnancy and lactation. The female offspring that were studied received a galactose-containing diet in the 3-week post-weaning period, which was followed by a Western-style diet mainly based on palm oil. The mice were sacrificed in a challenged state to enhance possible effects on insulin signaling. While the study design was identical to the previous study [17], there were dietary alterations unrelated to the focus of the intervention (glucose and galactose were identical). These changes were intended to mimic unintended changes as they may occur between different labs. In the chow breeding diet (rather than a semi-synthetic breeding diet), more starch—rather than some maltodextrin in the programming diet—and more unhealthy, saturated fatty acids containing HFD were used. Programming effects on obesity and associated metabolic parameters were investigated, including a focused analysis of the WAT-secreted insulin sensitizer adiponectin on muscle signaling.

## 2. Results

### 2.1. Body Weight, Body Composition, and Food Intake Development

During the three-week post-weaning period, mean body weight, lean mass, and fat mass increased significantly over time in both the control group (GLU; 32 en% glucose) and intervention group (GAL; 16 en% glucose and 16 en% galactose), as expected (Figure 1A–C), without differences between both dietary groups. Remarkably, mean food intake was significantly higher in the mice fed the GAL diet (*p* < 0.01) (Figure 1D).

During the following nine-week HFD period, body weight, fat mass, and lean mass increased significantly over time in both GLU and GAL mice (Figure 1A–C), without differences between both dietary groups. Food intake was no longer different between the two groups during the HFD period (Figure 1D); the drop in food intake at 14 weeks of age (week 8 of the HFD period) was the result of the fasting and refeeding challenge given in that week. Water intake was previously shown to increase by galactose versus glucose in another subset of mice [18]; here, in the early phase of the HFD period, water intake, 3.0 ± 0.4 vs. 3.2 ± 0.6 (GLU vs. GAL, mean ± SD, g/day, *p* = 0.44), was also not affected by the early life post-weaning diet, suggesting that increased water intake relates to the presence of dietary galactose specifically.

### 2.2. Oral Glucose Tolerance Test

After five weeks on the HFD, an oral glucose tolerance test (OGTT) was performed to assess glucose tolerance in the then 11-week-old mice (Figure 2). Fasting basal glucose levels were similar, and glucose tolerance was not significantly different between GLU and GAL mice, either by the individual time point (Figure 2A) or by the incremental area under the curve (iAUC) of glucose (Figure 2B). Plasma insulin concentrations were not significantly different between GLU and GAL mice at all three time points (Figure 2C) but insulin iAUC suggested that GAL mice had a lower insulin sensitivity (*p* = 0.03) (Figure 2D). However, the HOMA-IR index, based on basal levels, indicated that insulin resistance was similar between the two groups (Figure 2E).

### 2.3. Indirect Calorimetry

To investigate (in more detail) the metabolism and substrate utilization of mice that were on the HFD for 7 weeks, mice were studied in an indirect calorimetry (Inca) system. Respiratory exchange ratio (RER) patterns are shown in Appendix A; there was no significant difference in the mean 24-h RER between GLU and GAL mice (Appendix A). Energy expenditure (EE) tended to be higher (*p* = 0.08) during the 24-h period (Appendix A); it was significantly higher in GAL mice during the dark phase (*p* = 0.01) (Appendix A) without differences during the light phase (Appendix A). Twenty-four-hour HFD food intake within Inca of the GAL mice was higher than the GLU mice (*p* = 0.03) (Appendix A), even though they showed similar food intake over the full nine-week HFD-feeding period (Figure 1D).

### 2.4. Organ Weights and Serum Parameters

Organ weights indicated that mean gonadal WAT (gWAT) mass and mesenteric WAT (mWAT) mass were similar in both groups (Table 1). The mean liver weight and hepatic triglyceride content were not affected by the post-weaning diet (Table 1). The mean pancreatic weight of GAL mice was significantly higher than that of GLU mice (*p* = 0.03), without effects on serum insulin levels 15 min after the glucose bolus (Table 1).

Serum adiponectin concentrations, known as insulin sensitizers secreted from WAT [19], were significantly higher in the GAL mice (*p* = 0.02) (Table 1). Serum glucose concentrations were not different between the two groups. Serum leptin levels, indicators for whole body WAT mass, were not different, in line with the total fat mass (Figure 1B) and individual WAT depot masses (Table 1). Serum-free fatty acids and insulin-like growth factor 1 (IGF1) concentrations were also not affected by the post-weaning diet (Table 1).

### 2.5. Gene Expression and Protein Phosphorylation

To explore differences associated with the higher serum adiponectin levels in GAL mice, adiponectin signaling was analyzed at the mRNA level in gWAT (Figure 3), liver (Figure 4), and skeletal muscles (Figure 5). First, the gene expression level of *Adipoq* mRNA in gWAT was examined, which indeed was significantly higher in GAL mice (*p* = 0.03), independent of *Rarres2* (chemerin) expression. (Figure 3). The liver and skeletal muscles are important target organs for adiponectin, with adiponectin receptors 1 and 2 serving as the predominant receptors for adiponectin and playing important roles in the regulation of glucose and lipid metabolism [20]. Hepatic expression levels of *AdipoR1*, *AdipoR2, Rarres2*, *Screbf1*, and *Acox1* mRNA were not significantly affected (Figure 4). In skeletal muscles, transcript levels of *Acacb* (a regulator of fatty acid synthesis) were significantly higher in GAL mice (*p* < 0.01), while *AdipoR1*, *AdipoR2*, *Rarres2*, *Pdk4*, *Cpt1b*, *Acly*, and *Pgc1a* were not significantly different between the two groups (Figure 5).

In addition, insulin signaling was studied, since it was previously shown that *Irs2* transcript levels in WAT were significantly reduced by metabolic programming by galactose versus glucose [17]. Expression levels in gWAT of *Irs1*, *Irs2*, and its downstream target *Tbc1d4* were similar in both groups (Figure 3), and normalized AKT Ser473 phosphorylation in gWAT also appeared to not be significantly different (Appendix A), cumulatively suggesting a lack of differential effects on insulin signaling in gWAT. Of note, expression levels of *Irs2* in skeletal muscles were significantly lower in GAL mice (*p* = 0.01), without differences in *Irs1* expression levels (Figure 5).

The gene expressions of inflammatory markers, including *Saa1* and *S100a8*, were explored in gWAT and the liver; none of the markers were affected in adulthood by the post-weaning diet given in early life (Figure 3 and Figure 4). Additionally, the analysis of gene expression in gWAT indicated that the expressions of several genes involved in lipolysis and WAT expandability (Figure 3) were not programmed by the post-weaning diet.

## 3. Discussion

In this study, the nutritional programming effects of the presence of dietary galactose in the immediate post-weaning period were investigated in female mice. Body weight, body composition, glucose tolerance, circulating parameters, and transcripts of different organs were studied in later life in an obesogenic environment. Partial replacement of glucose with galactose in an early life post-weaning diet for only three weeks followed by a high-fat diet for nine weeks significantly decreased *Irs2* expression in skeletal muscle. Moreover, dietary galactose significantly increased *Adipoq* expression in gWAT and serum adiponectin levels in later life without differences in body weight and body composition between the two groups. Glucose tolerance, and in support, serum insulin levels, and insulin signaling in gWAT were all not affected. Similarly, the expressions of inflammatory genes in the liver and gWAT also appeared to be unaltered.

As the total fat mass at the end of this study was similar between groups, this contrasts with a previous study based on the same nutritional intervention by galactose versus glucose [17], although the same study design and the same mouse strain were used. However, the studies differed with regard to dietary aspects unrelated to the focus of the intervention. First, in this study, we used a chow diet during breeding, while in the previous study [17], the breeding diet was a semi-synthetic diet. Second, the carbohydrate composition of the post-weaning (intervention) diet we used here was based on 28 en% wheat starch and about 3 en% fructose was used to soften the food pellet, while previously, the carbohydrate fraction was composed of 16 en% wheat starch and 16 en% maltodextrin [17]. Lastly, here, the HFD was based mainly on palm oil and, thus, was rich in saturated fatty acids, while previously, the HFD was based on the BIOCLAIMS HFD, with a healthy, unsaturated fatty acids profile [21,22]. The dietary differences during breeding and post-weaning periods may possibly have caused the different programming effects. Moreover, given that the HFD in this study was more adverse when given for 26 weeks to adult mice [23], possibly, the more adverse high-fat diet used in this study compared with the diet used in our previous study [17], prevented or suppressed potential beneficial programming effects of dietary galactose given in the post-weaning period. Although we hypothesized that the galactose programming effects would be identical and fully resistant to the imposed differences, this was not the case. While we again show post-weaning galactose programming effects, further underpinning potential robustness, the detailed molecular responses differ. This has important implications for the reporting of programming studies as evidently, details matter. In addition, this urges follow-up research on underlying mechanisms.

Due to HFD-induced obesity, metabolic dysfunction, such as insulin resistance, is seen regularly [24]. In our study, the OGTT results indicated that the post-weaning diet followed by five weeks of HFD-feeding had no effects on glucose tolerance, except on the insulin iAUC. In light of the similarities seen in basal fasting serum levels of insulin and glucose (Figure 2), and the similar serum insulin levels and glucose levels at the end of the study at PN105, 15 min after a glucose challenge (Table 1), the overall interpretation is that these measures indicate a similar glucose tolerance and insulin sensitivity between the intervention groups. Of note, the galactose/glucose intervention was only during the first three weeks after weaning, followed by the same HFD in both intervention groups thereafter. In contrast, multiple studies have measured the direct effects of glucose and galactose on insulin release. For example, in humans, a galactose drink resulted in a lower increase in circulating insulin concentrations than a glucose drink [25,26,27,28]. In addition, a drink with only galactose resulted in a lower insulin increase than a drink with glucose and galactose (1:1) [29,30]. Moreover, in the long-term, feeding rats a diet containing 15 en% galactose compared to 15 en% glucose for nine weeks improved hepatic insulin sensitivity in adult males [31]. Indeed, in support, fasting serum insulin levels at the end of the post-weaning intervention period at PN42 were lower in the GAL mice [18]. However, apparently after an additional nine weeks of HFD-feeding, the differences were dampened. We also examined the insulin signaling-related gene expression in gWAT and skeletal muscles. Insulin receptor substrate proteins play vital roles in insulin signaling, with *Irs1* and *Irs2* representing the most important ones [32]. *Irs2* expression in the skeletal muscle was significantly lower in GAL mice (Figure 5), while no differences were seen in *Irs1* expression (Figure 5), and both receptors showed no differential expression in gWAT (Figure 3). However, in humans, it has been reported that in the subcutaneous WAT of insulin-resistant versus insulin-sensitive women, *Irs2* appeared to be lower-expressed [33]. Together with the previous galactose nutritional programming study [17], the differential expression of *Irs2* seems to be consistently affected, although in different tissues. How this translates into future beneficial health effects remains to be established.

Obesity is associated with low-grade chronic inflammation, which is a key factor underlying insulin resistance [34]. Therefore, transcript levels of inflammatory markers in the liver and gWAT were measured. Previously, we showed that the three-week post-weaning dietary galactose intervention reduced hepatic expression of inflammatory markers [18]. Here, the expression levels of inflammatory markers were not different in gWAT (Figure 3) or the liver (Figure 4) after a HFD period. Early life galactose did not program the beneficial effects on inflammation in later life. Possibly, the previously observed beneficial effects on inflammation in early life might be depressed by later-life obesity-induced proinflammatory responses. However, the mice were still relatively young even at the end of the study, and their liver and WAT depots were evidently not inflamed. It would be of interest to examine how the liver and WAT inflammation develops late in life, and whether programmed differences emerge at, e.g., 15 or 18 months, which corresponds to the age where low-grade liver and adipose tissue dysfunctions surge in humans. This remains subject to future investigations.

IGF1 is mainly produced by the liver [35]. Adiponectin is mainly synthesized in adipocytes, and serum adiponectin levels are generally lower in obese individuals; there is a functional interplay between IGF1 and adiponectin [36]. Although no differences were seen in serum IGF1 levels and fat mass, the serum adiponectin levels (Table 1) and gWAT *Adipoq* expression (Figure 3) were significantly higher in the GAL mice. This is in line with another nutritional programming study on mice, which showed that maternal overweight significantly reduced high-molecular-weight adiponectin in the offspring, also without changes in fat mass [37]. Chemerin and adiponectin are known as contrary adipokines [38,39,40]. Chemerin is encoded by gene *Rarres2*, which is highly expressed in multiple organs in the body. In this study, we did not observe any significant difference in the expression of *Rarres2* between the groups in gWAT, the liver, or EDL muscle (Figure 3, Figure 4 and Figure 5). Consequently, the expressions of *Adipoq* and *Rarres2* were not significantly correlated in gWAT, implicating that the increase in adiponectin is not paralleled by a decrease in chemerin. Adiponectin receptors 1 and 2 serve as receptors for adiponectin, and they play key roles in insulin sensitivity by the activation of AMP-activated protein kinase (AMPK) and peroxisome proliferator-activated receptor (PPAR)-α [41]. The adiponectin receptors and downstream targets of AMPK and PPAR pathways in the liver and skeletal muscle were explored. No significant differences were found in any of the analyzed genes/proteins (Figure 4, Figure 5, and Appendix A). Overall, although it is tempting to speculate that galactose in post-weaning diets improved metabolic health in later life by increasing circulating adiponectin levels, further research is needed to elucidate the mechanisms and actions of increased serum adiponectin levels and/or adiponectin signaling.

In conclusion, partial replacement of glucose with galactose in an early life post-weaning diet for only three weeks followed by a high-fat diet for nine weeks significantly increased white adipose tissue adiponectin gene expression and serum adiponectin levels in later life without alterations in fat mass, inflammation status, and glucose tolerance, together with lower skeletal muscle *Irs2* expression.

## 4. Materials and Methods

### 4.1. Diets

Two semi-synthetic post-weaning diets and one semi-synthetic high-fat diet were ordered from Research Diet Services BV (Wijk bij Duurstede, The Netherlands). Post-weaning diets were composed in accordance with AIN-93 guidelines for growing rodents [42]. A detailed composition can be found in Appendix A. The macronutrient composition of the post-weaning diets was 20 en% protein, 16 en% fat, and 64 en% carbohydrates. The carbohydrates in the post-weaning diets were mainly wheat starch (28 en%) and monosaccharides (32 en%), i.e., only 32 en% glucose in the glucose diet, and a glucose and galactose combination (in 1:1 ratio; 16 en% each) in the glucose + galactose diet (Appendix A). The HFD contained 20 en% protein, 40 en% fat (mainly palm oil-based), and 40 en% carbohydrates (Appendix A). This type of HFD was shown to induce a more adverse metabolic state than BIOCLAIMS HFD after long-term feeding [18,23] and corresponds to the human macronutrient intake profile. To all diets, vitamin and mineral mixes were added to ensure that nutrient recommendations were met [42].

### 4.2. Animals and Ethical Approval

Ethical approval for the animal experiments and procedures was granted by national and local animal experimental committees (AVD1040020171668; 2017.W-0024.001). Experiments were executed following the EU directive 2010/63/EU. All experiments were carried out at 23 ± 1 °C, with a 12:12 light–dark cycle, and ad libitum access to food, unless stated otherwise. Breeding pairs (C57BL/6JRccHsd mice) were ordered from Envigo (Horst, The Netherlands). After an adaptation period of two to five weeks in the animal facilities, mice were time-mated. At postnatal days 1 or 2, nests were cross-fostered and standardized to 6 pups per nest, with 2–4 females per nest. During adaptation, pregnancy, and lactation, dams were fed a standard breeding chow (AM-II, AB Diets, Woerden, The Netherlands). On day 21 (±1), female pups were stratified by body weight and fat mass, as determined by EchoMRI 100V (EchoMedical Systems, Houston, TX, USA). One group was assigned to the GLU diet (n = 26), and the other group to the GAL diet (n = 26). The mice were housed individually. After three weeks on these diets, subsets (n = 14) were switched to the same HFD for nine weeks. The remaining mice (GLU, n = 12 and GAL, n = 12) were killed on day 42 for the molecular evaluation of direct effects, and these have been reported previously [18]. As a result, here, we focus only on the subsets continuing on the HFD (n = 14), with the respective groups being referred to as either GLU or GAL, depending on the postweaning intervention diet.

### 4.3. Experimental Setup and Measurements

Body weight was measured weekly. Body composition, consisting of fat mass and lean mass, was measured weekly during the three-week post-weaning period, and biweekly throughout the HFD period. Food intake was determined weekly by subtracting the weight of remaining food pellets from the weight of the food pellets provided. Some mice on the HFD crumbled pellets without eating everything, causing food crumbs to be dispersed within the bedding, and making food intake quantification impossible. Therefore, 5 mice from the GLU diet group and 2 mice from the GAL diet group were excluded from the regular HFD food intake measurements. Water intake (72 h) was measured in a subset of mice by subtracting the weight of the water bottle on day 53 from day 50 (n = 7 mice per group). Indirect calorimetry measurements were performed (n = 12 mice per group) from day 93 to day 98 (See below).

### 4.4. OGTT

On day 77, an oral glucose tolerance test was performed. Mice were fasted for five hours in a light phase. A tail cut was made for whole blood glucose measurements and the plasma collection. A glucose bolus (2.0 g/kg body weight) was given by oral gavage. Blood glucose was measured with a freestyle blood glucose system (Abbott Diabetes Care, Hoofddorp, The Netherlands) at time points t = 0, 15, 30, 45, 60, 90, and 120 min. Blood plasma was collected using Microvette tubes (Starstedt, Nümbrecht, Germany) at time points t = 0, 15, and 30 min; samples were centrifuged for 5 min at 2000× *g*, and plasma was stored at −80 °C until further analysis. One of the GLU mice died unexpectedly during this procedure, and the data of this individual mouse were excluded from the start of the HFD-feeding (day 42) onwards for all measurements.

### 4.5. Indirect Calorimetry

Inca measurements were performed from day 93 to day 98. Measurements of EE, RER, and food intake were done as described previously [43] (Phenomaster LabMaster Metabolism Research Platform, TSE systems GmbH, Bad Homburg, Germany). The first day in the Inca was considered the adaptation period, and the next 24 h (one light phase and one dark phase) were analyzed as the experimental period. This was followed by a fasting-and-refeeding challenge to assess metabolic flexibility in vivo, as published [15]. As not all food provided was consumed at once during the fasting and refeeding challenge, these data were not further analyzed.

### 4.6. Dissection

On day 105, mice were food-deprived at the start of the light phase for 2–4 h. Fifteen minutes before sacrifice, mice were given a glucose bolus (2.0 g/kg body weight) by oral gavage, to have them in a challenged state with peaking insulin levels, upon sacrifice. Mice were decapitated, and whole blood glucose levels were immediately measured (Freestyle, Abbott Diabetes Care, Hoofddorp, The Netherlands). The remainder of the blood was collected in MiniCollect serum tubes (Greiner Bio one B.V., Alphen aan de Rijn, The Netherlands), centrifuged for 10 min at 3000× *g* at 4 °C, and the obtained serum was aliquoted and stored at −80 °C. Liver, gWAT depot, and skeletal muscle (extensor digitorum longus muscle) tissue were collected, snap-frozen in liquid nitrogen, and stored at −80 °C until further analysis.

### 4.7. Serum and Plasma Analyses

Serum leptin and adiponectin concentrations were determined with Bio-Plex Pro Mouse diabetes assays (Bio-Rad laboratories, Veenendaal, The Netherlands), serum IGF1 with the mouse IGF1 kit (R&D system, Wiesbaden Nordenstadt, Germany), and serum free fatty acids with the NEFA-HR(s) kit (Fujifilm Wako Chemicals Europe GMBH, Neuss, Germany). Serum insulin (at sacrifice) and plasma insulin (during OGTT) concentrations were measured with an Ultra-Sensitive Mouse Insulin ELISA Kit (Crystal Chem, Zaandam, The Netherlands). All assays were performed according to the manufacturers’ instructions. Samples were tested in duplicate, except for the plasma insulin measurements of the OGTT due to the limited amount of samples available. A HOMA-IR index was calculated with the C57BL/6J mouse-specific formula [44], using whole blood fasting glucose and plasma insulin concentrations at t = 0 from the OGTT.

### 4.8. Hepatic Triglycerides Content

Hepatic triglycerides content was measured using the Triglycerides Liquicolormono kit (HUMAN, Wiesbaden, Germany). Protein content was measured with the DC protein assay (Bio-Rad laboratories, Veenendaal, The Netherlands). All assays were performed according to the manufacturers’ instructions. Samples were tested in duplicate.

### 4.9. Gene Expression

Total RNA was extracted from gWAT and liver using RNeasy columns (Qiagen, Venlo, The Netherlands), and was extracted from skeletal muscle using Invitrogen TRIzol reagent (Thermo Fisher Scientific, Schwerte, Germany), according to the manufacturer’s instructions. RNA integrity was verified on the Agilent 2200 Tapestation (Agilent Technologies Inc, Santa Clara, CA, USA). cDNA synthesis and real-time quantitative reverse transcription polymerase chain reaction were executed as described [17] using beta-2 microglobulin (*B2m*), ribosomal protein S15 (*RPS15*), and calnexin (*Canx*) as reference genes. Primer sequences and annealing temperatures can be found in Appendix A. Data are expressed as fold-change to the mean of the GLU mice.

### 4.10. Western Blot

Western blot for phospho-AKT (Ser473), total AKT, and β-actin were performed on gWAT protein extracts as described [45]. Briefly, the protein concentration was measured using the DC protein assay (Bio-Rad laboratories, Veenendaal, The Netherlands), and 20 µg of a protein sample was run on a 12% acrylamide gel and blotted on a PVDF membrane (Merck Millipore, Amsterdam, The Netherlands). The membrane was incubated with primary antibodies (the antibody against β-actin was purchased from Abcam, Cambridge, UK; antibodies against AKT and phospho-AKT (Ser473) were purchased from Cell Signaling Technology, via BIOKÉ, Leiden, The Netherlands) at 4 °C overnight and then was incubated with goat anti-mouse secondary antibody for β-actin or donkey anti-rabbit secondary antibody (LI-COR, Lincoln, NE, USA) for the other antibodies, all at room temperature for 1 h. The membrane was scanned on an Odyssey scanner (LI-COR, Lincoln, NE, USA). Bands were analyzed using Odyssey software V3.0 (LI-COR, Lincoln, NE, USA).

### 4.11. Statistics

Data were analyzed with GraphPad Prism, version 9 (GraphPad Software, Inc., San Diego, CA, USA). Two-way ANOVA was used for the analyses of body weight, fat mass, and lean mass, with the post-weaning diet as the between-subject factor, time as the within-subject factor, and the group x time interaction. Post-weaning and HFD periods were studied separately. OGTT data and RER pattern data were also analyzed with two-way ANOVA. Other parameters were analyzed with a Students’ unpaired two-tailed *t*-test for normally distributed data, a *t*-test, with the Welch correction for normally distributed data with unequal variances, or a Mann–Whitney U for not normally distributed data. D’Agostino–Pearson omnibus normality tests were used to test for normality; data were log-transformed when the original distributions were not normal, and retested for normality. Results are given as mean ± SD unless stated otherwise; *p*-values < 0.05 were considered significant.

## Figures and Tables

**Figure 1 ijms-23-10207-f001:**
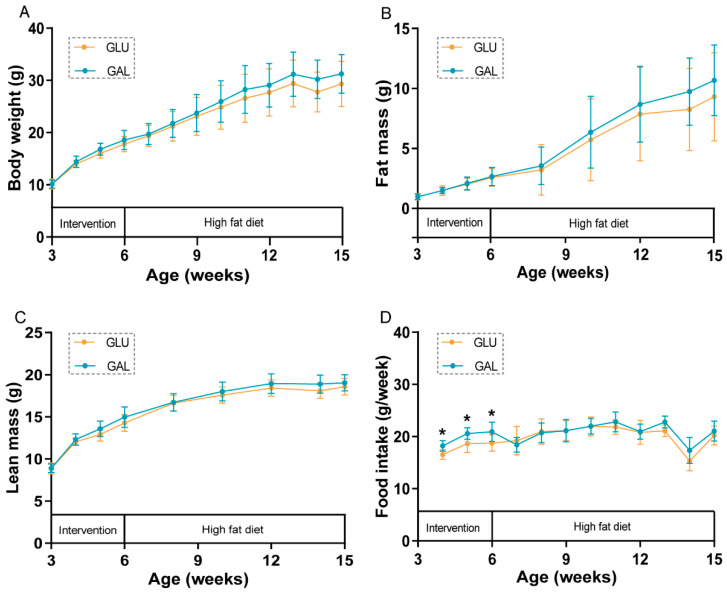
Body weight and body composition development, and food intake in GLU and GAL female mice. (**A**) Body weight, (**B**) fat mass, (**C**) lean mass, and (**D**) food intake development during the intervention (aged 3–6 weeks) and consecutive high-fat diet (HFD) (aged 6–15 weeks) periods. Values represent mean ± SD, n = 13–14. * *p* < 0.05.

**Figure 2 ijms-23-10207-f002:**
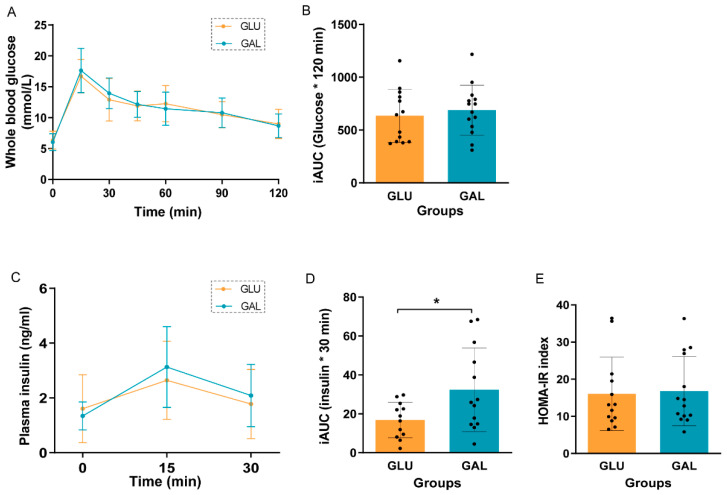
Oral glucose tolerance test (OGTT) in GLU and GAL mice after five weeks on a HFD. (**A**) Blood glucose curves over time after a 2.0 g/kg body weight gavage, (**B**) incremental area under the curve (iAUC) of the glucose curve, (**C**) plasma insulin concentrations in the first 30 min of OGTT, (**D**) iAUC of the insulin curve, and (**E**) homeostatic model assessment for the insulin resistance (HOMA-IR) index. Values are given as mean ± SD, n = 12–14. * *p* < 0.05.

**Figure 3 ijms-23-10207-f003:**
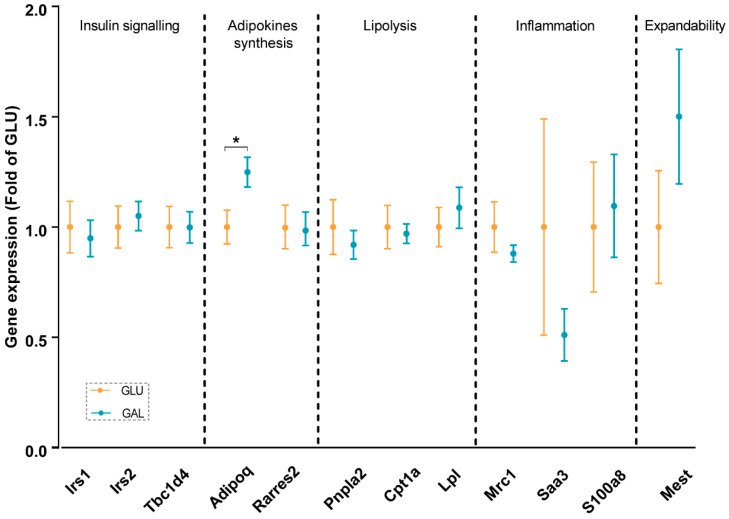
Gonadal white adipose tissue gene expression in GLU and GAL mice followed by nine weeks of a HFD. *Irs1*: insulin receptor substrate 1; *Irs2*: insulin receptor substrate 2; *Tbc1d4*: TBC1 domain family member 4; *Adipoq*: adiponectin, C1Q, and collagen domain containing; *Rarres2*: retinoic acid receptor responder 2; *Pnpla2*: patatin-like phospholipase domain containing 2; *Cpt1a*: carnitine palmitoyltransferase I; *Lpl*: lipoprotein lipase; *Mrc1*: mannose receptor C type 1; *Saa3*: serum amyloid A 3; *S100a8*: S100 calcium-binding protein A8; *Mest*: mesoderm-specific transcript homolog protein. Values are expressed as mean ± SEM (n = 10–14). * *p* < 0.05.

**Figure 4 ijms-23-10207-f004:**
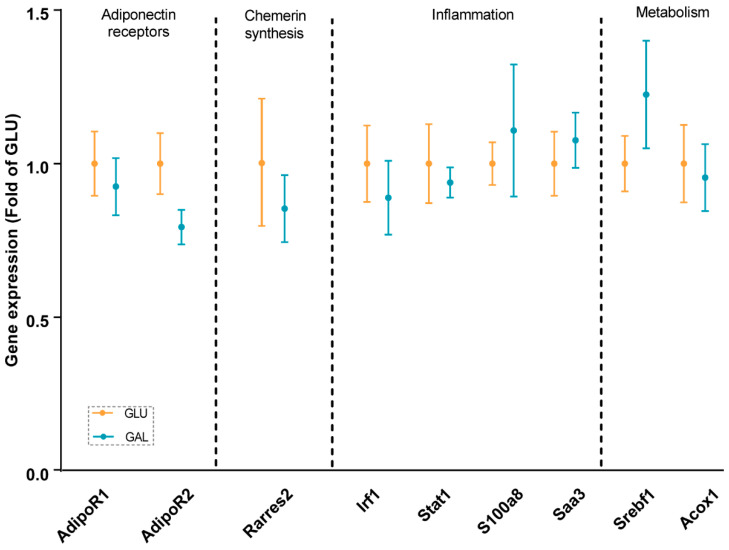
Hepatic gene expression in GLU and GAL mice followed by nine weeks of a HFD. *AdipoR1*: adiponectin receptor 1; *AdipoR2*: adiponectin receptor 2; *Rarres2*: retinoic acid receptor responder 2; *Irf1*: interferon regulatory factor 1; *Stat1*: signal transducer and activator of transcription 1; *S100a8*: S100 calcium binding protein A8; *Saa3*: serum amyloid A 3; *Srebf1*: sterol regulatory element binding transcription factor 1; *Acox1*: acyl-CoA oxidase 1. Values are expressed as mean ± SEM (n = 7–12).

**Figure 5 ijms-23-10207-f005:**
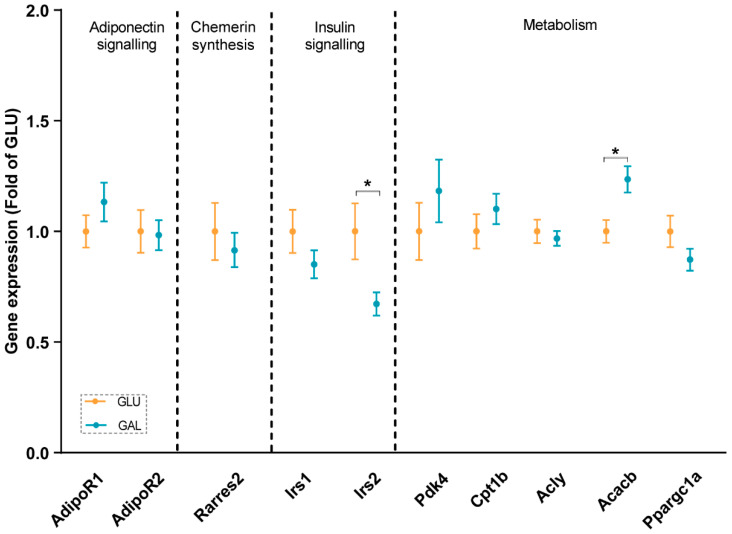
Skeletal muscle gene expressions in GLU and GAL mice followed by nine weeks of a HFD feeding. *AdipoR1*: adiponectin receptor 1; *AdipoR2*: adiponectin receptor 2; *Rarres2*: retinoic acid receptor responder 2; *Irs1:* insulin receptor substrate 1; *Irs2:* insulin receptor substrate 2; *Pdk4*: pyruvate dehydrogenase kinase 4; *Cpt1b*: carnitine palmitoyltransferase 1b; *Acly*: ATP citrate lyase; *Acacb*: acetyl-CoA carboxylase beta; *Ppargc1a*: peroxisome proliferative activated receptor, gamma, coactivator 1 alpha. Values are expressed as mean ± SEM (n = 13–14). * *p* < 0.05.

**Table 1 ijms-23-10207-t001:** Organ weights and serum parameters in GLU and GAL mice at the end of the study ^1^.

Items	GLU	GAL
Liver weight (g)	1.22 + 0.43	1.19 + 0.24
Gonadal white adipose tissue weight (g)	0.62 ± 0.27	0.70 ± 0.23
Mesenteric white adipose tissue weight (g)	0.59 ± 0.29	0.67 ± 0.21
Pancreas weight (g)	0.25 ± 0.03	0.29 ± 0.07 *
Liver triglycerides (µg/mg protein)	148.3 ± 19.3	152.9 ± 24.1
Blood glucose (mmol/L)	10.6 ± 3.6	10.4 ± 2.7
Serum insulin (ng/mL)	2.37 ± 1.35	2.56 ± 1.08
Serum adiponectin (µg/mL)	10.1 ± 1.2	11.6 ± 2.12 *
Serum insulin-like growth factor 1 (µg/mL)	299.5 ±161.6	299.4 ± 164.5
Serum leptin (ng/mL)	29.9 ± 6.5	38.0 ± 5.6
Serum-free fatty acids (mmol/L)	0.57 ± 0.02	0.59 ± 0.03

^1^ After three weeks of post-weaning intervention (GLU or GAL diet) at 6 weeks of age, all mice received the same HFD for 9 weeks. Fifteen minutes prior to sacrifice, food-deprived mice received a glucose bolus. Values are given as mean ± SD, n = 13–14. * *p* < 0.05.

## Data Availability

The data presented in this study are available upon request.

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
