# Peer review of "Galactose in the Post-Weaning Diet Programs Improved Circulating Adiponectin Concentrations and Skeletal Muscle Insulin Signaling"

_ijms, 2022, doi:10.3390/ijms231810207_

Round 1
Reviewer 1 Report
The Sun et al., 2022, Manuscript ID: ijms- 1849842 addresses how galactose in the post-weaning diet programs improved circulating adiponectin levels and skeletal muscle insulin signaling. A search on Pubmed.gov for the terms "galactose" and "adiponectin” and “Skeleton muscle” keywords resulted in no hits that depicts the novelty of this study. There are few important queries and few suggestion which makes this manuscript more representable to be publish.
1. If it is possible can the author provide protein expression of AdipoR1/2 in the skeleton muscle of glucose and galactose feed mice? It will give us clear picture that the increased circulating AdipoQ level is acting on the skeleton muscle or somewhere else? If they have the tissue kindly check in the liver tissue also the expression of AdipoR1/2 proteins as it is the main metabolic organ.
2. If possible, can the authors show the histomorphological changes in the skeleton muscle glucose and galactose fed mice?
3. If possible can you check the level of contrary adipokine chemerin in the study to give it more strength and cite the MS “Adiponectin and Chemerin: Contrary Adipokines in Regulating Reproduction and Metabolic Disorders”?
Reviewer 2 Report
this is a manuscript reporting the findings of an original study surveying the possible benefits of "galactose" in adulthood glucose metabolism. this study provides fairly novel data and is well-structured. the authors provide evidence demonstrating galactose therapy improves glucose homeostasis and modulates adiponectines levels. but in my point view, this is not at importance since is not functional for clinical use. however, based on Editor in Chief final decision, can be acceptable.
Round 2
Reviewer 1 Report
The authors have tried to justify all the queries raised by me.